# Machine Learning for Predicting the Low Risk of Postoperative Pancreatic Fistula After Pancreaticoduodenectomy: Toward a Dynamic and Personalized Postoperative Management Strategy

**DOI:** 10.3390/cancers17111846

**Published:** 2025-05-31

**Authors:** Roberto Cammarata, Filippo Ruffini, Alberto Catamerò, Gennaro Melone, Gianluca Costa, Silvia Angeletti, Federico Seghetti, Vincenzo La Vaccara, Roberto Coppola, Paolo Soda, Valerio Guarrasi, Damiano Caputo

**Affiliations:** 1Operative Research Unit of General Surgery, Fondazione Policlinico Universitario Campus Bio-Medico, 00128 Rome, Italy; v.lavaccara@policlinicocampus.it (V.L.V.); r.coppola@policlinicocampus.it (R.C.); d.caputo@policlinicocampus.it (D.C.); 2Unit of Computer Systems & Bioinformatics, Department of Engineering, Università Campus Bio-Medico di Roma, 00128 Rome, Italy; filippo.ruffini@unicampus.it (F.R.); p.soda@unicampus.it (P.S.); valerio.guarrasi@unicampus.it (V.G.); 3Università Campus Bio-Medico di Roma, 00128 Rome, Italy; alberto.catamero@unicampus.it (A.C.); gennaro.melone@unicampus.it (G.M.); federico.seghetti@alcampus.it (F.S.); 4Department of Life Science, Health, and Health Professions, Link Campus University, 00165 Rome, Italy; g.costa@policlinicocampus.it; 5Unit of Laboratory, Fondazione Policlinico Universitario Campus Bio-Medico, 00128 Rome, Italy; s.angeletti@policlinicocampus.it; 6Research Unit of Clinical Laboratory Science, Department of Medicine and Surgery, Università Campus Bio-Medico di Roma, 00128 Rome, Italy; 7Research Unit of General Surgery, Università Campus Bio-Medico di Roma, 00128 Rome, Italy

**Keywords:** pancreatic ductal adenocarcinoma (PDAC), machine learning (ML), postoperative pancreatic fistula (POPF)

## Abstract

Postoperative pancreatic fistula remains one of the greatest challenges regarding the management of patients who underwent pancreatic surgery, increasing the incidence of secondary complications and prolonging hospital stays. Thus, the development of an accurate model for stratifying the risk of pancreatic fistula is currently a priority, but it still presents significant challenges. Machine learning algorithms provide a real opportunity to refine the risk stratification models, due to their ability to process and analyze complex variables and large datasets and to dynamically search for correlations among them. This article contributes to the existing literature by exploring the application of machine learning in refining the risk stratification of pancreatic fistula, offering a highly accurate prediction model, in order to improve patient management strategies.

## 1. Introduction

Pancreaticoduodenectomy (PD) represents the main upfront strategy for resectable pancreatic ductal adenocarcinoma (PDAC) and periampullary tumors, as well as for borderline resectable and locally advanced PDAC after downstaging following neoadjuvant treatment [1]. However, the clinical and oncological outcomes of this procedure are influenced by the high incidence of postoperative complications, which occur in approximately 50% of cases [2,3].

Postoperative pancreatic fistula (POPF) represents the most significant complication due to its complex management and its potential consequences on short-term surgical outcomes, as well as on subsequent access to key adjuvant treatments for an effective comprehensive oncologic strategy [4]. The management of POPF has recently shifted from a standardized and uniform approach that did not account for individual patient differences to a more proactive strategy focused on early prediction and personalized treatment [5]. This new approach leverages predictive models to improve prevention and guide tailored clinical decisions. While previous systems provided a general estimate of POPF incidence and were appreciated for their simplicity, their predictive accuracy remains uncertain [6]. Further research is needed to develop more precise and reliable models [7].

Machine learning (ML), a branch of artificial intelligence (AI), has become an integral tool in various fields, including medical science, where it is revolutionizing diagnostics, treatment planning, and outcome prediction [8]. ML models, structured through multiple processing layers, are capable of learning complex patterns from vast datasets, enabling them to identify intricate relationships that may not be immediately apparent through traditional statistical methods [9]. In the surgical field, ML is being increasingly utilized for risk assessment, intraoperative guidance, postoperative complication prediction, and personalized patient management [10]. Beyond surgery, ML is also widely applied in pharmacogenomics, where it aids in predicting drug responses based on genetic profiles, as well as in medical imaging, improving the accuracy of disease classification and detection. Furthermore, ML-driven decision support systems are enhancing clinical workflows by integrating vast amounts of patient data to assist clinicians in making more informed, evidence-based decisions [11].

Considering the potential of AI in this field, our study aims to develop an innovative predictive strategy to identify patients at low risk of clinically relevant postoperative pancreatic fistula (POPF) after pancreaticoduodenectomy (PD) using machine learning (ML) algorithms. By analyzing an extensive set of preoperative, intraoperative, and early postoperative variables, all obtainable within the first three postoperative days, our goal is to achieve greater predictive accuracy compared to traditional models, based on linear combinations of input data (e.g., logistic regression). A more precise risk assessment could enable more personalized clinical management, guiding early interventions such as timely drain removal and optimizing postoperative care. This approach has the potential to significantly improve patient outcomes, refine surgical decision-making, and ultimately reduce the burden of complications like POPF.

In this work, we addressed the prediction of postoperative pancreatic fistula (POPF) through a comprehensive machine learning framework. A total of twenty-four ML algorithms were implemented and evaluated using a wide array of preoperative, intraoperative, and early postoperative variables. Model performance was assessed using both the Matthews Correlation Coefficient (MCC) and AUC-ROC, with model selection tailored to the specific prediction task, based on the best-performing metric—an approach that allows adaptive and context-sensitive decision-making.

In addition to comparing multiple ML models, we included logistic regression as a traditional multivariate baseline, enabling a direct assessment of the added value of non-linear ML methods. Finally, we applied explainable AI (XAI) techniques, specifically SHAP (SHapley Additive exPlanations), to enhance interpretability and identify clinically relevant predictors in the early postoperative window.

The main contributions of this study are:-We conducted a large-scale comparative analysis involving twenty-four machine learning algorithms to predict POPF after pancreaticoduodenectomy, leveraging a structured and temporally aligned dataset including preoperative, intraoperative, and early postoperative variables.-We implemented a task-specific model selection strategy based on the performance of each algorithm using the Matthews Correlation Coefficient (MCC), a metric particularly well-suited for imbalanced clinical classification problems.-We included a classical multivariate logistic regression model as a comparator to assess the added value of modern ML approaches. This allowed us to demonstrate the performance gains offered by non-linear, ensemble-based methods such as GradientBoostingClassifier, which consistently outperformed the baseline in multiple tasks.-We integrated explainable AI (XAI) techniques, specifically SHAP (SHapley Additive exPlanations), to provide interpretability of model outputs. This enabled the identification of the most influential clinical features contributing to model decisions, enhancing transparency and supporting clinical insight generation.-We focused on the early prediction of POPF risk within the first three postoperative days, aligning our modeling approach with real-world clinical decision points, such as the timing of drain removal.

This approach aims to support individualized postoperative management and promote trust in ML-assisted decision-making through transparency and model explainability.

## 2. Materials and Methods

### 2.1. Study Design, Population, and Variables

Patients who underwent surgery for periampullary cancer were retrospectively evaluated at the Division of General Surgery, Fondazione Policlinico Universitario Campus Bio-Medico, Rome, Italy, from January 2015 and January 2022. Consecutive patients were retrieved from a prospectively maintained institutional database. The following exclusion criteria were applied: (1) patients who underwent total pancreatectomy or distal pancreatectomy; (2) those with missing data. This study was approved by the institutional ethics committee of Fondazione Policlinico Universitario Campus Bio-Medico (approval no.: 2022.221) as a retrospective analysis of collected data, conducted in accordance with the ethical standards of the World Medical Association’s Declaration of Helsinki. Individual written consent for this retrospective analysis was waived.

Baseline patient parameters, including age, sex, body mass index (BMI), history of smoking, hypertension, cardiovascular and pulmonary comorbidities, diabetes mellitus, and American Society of Anesthesiology (ASA) score, were collected. These were the preoperative and intraoperative data encompassed: histological diagnoses, application of neoadjuvant therapy, serum amylase and lipase, bile culture, the surgical approach (minimally invasive or open pancreaticoduodenectomy), type of surgical procedure (i.e., pylorus-preserving pancreaticoduodenectomy or pylorus resecting pancreaticoduodenectomy), venous resection, pancreatic texture, Wirsung duct diameter, operating time, intraoperative blood loss. To investigate the impact of postoperative parameters on POPF, the laboratory values of serum lipase, DFA, and procalcitonin within the first three days after PD were collected.

### 2.2. Outcome and POPF Definition

The primary outcome was to develop a ML model to predict POPF following PD. The POPF was described in accordance with the updated ISGPS 2016 consensus guidelines [12]. Specifically, Grade A POPF was defined as “biochemical leak”, which is characterized by the amylase level, of drain fluid on postoperative day three or later of more than three times the upper reference value in serum. Grade B POPF was characterized by the presence of a postoperative pancreatic fistula along with at least one of the following conditions: (1) placement of a postoperative percutaneous drain; (2) a drain remaining in place for more than 21 days; (3) organ space infection or surgical site infection (SSI) without organ failure; (4) POPF-related hemorrhage or pseudo-aneurysm requiring an angiographic procedure or blood transfusion; (5) a clinically significant change in the management of POPF, such as artificial nutrition, prolonged hospitalization, or readmission. Grade C POPF was defined as the presence of a postoperative pancreatic fistula along with at least one of the following criteria: (1) reoperation; (2) organ failure; (3) death.

### 2.3. Statistical Analysis and Machine Learning Procedure

Baseline, preoperative, intraoperative, and postoperative characteristics were described. Continuous numerical variables did not follow a normal distribution across the overall population and within each POPF subgroup, so were expressed as median [interquartile range]. Dichotomous variables were noted as numbers and percentages. The Kruskal–Wallis test was used to compare groups and examine potential correlations between these variables and POPF occurrence.

For categorical variables, Pearson’s chi-squared test was used to assess group differences (Appendix A). A *p*-value of <0.05 was considered statistically significant.

### 2.4. Data Preprocessing

The data underwent a comprehensive preprocessing pipeline to ensure the quality and consistency of the data fed into the machine learning models. Numerical features were standardized using Min-Max scaling, transforming their values to a uniform range between −1 and 1. This scaling technique preserves the relationships among the data points while ensuring that no single feature dominates due to scale differences. Categorical variables were transformed into dummy variables through one-hot encoding. This process converted categorical labels into a binary matrix representation, allowing the algorithms to process these variables effectively without imposing any ordinal relationship.

We also addressed missing data, which accounted an overall missingness of 3.3%. As the missingness was moderate and plausibly consistent with a Missing At Random (MAR) mechanism, we employed K-Nearest Neighbor (KNN) imputation, which preserves local data structure and multivariate relationships, making it well-suited for clinical datasets. This imputation strategy allowed us to retain the full dataset and reduce potential bias in downstream model training.

To prevent data leakage and enhance the generalizability of the models, each dataset was randomly shuffled before splitting. An 80:20 train-test split was employed using the hold-out method, where 80% of the data were allocated for training and 20% for testing. For all tasks, stratified sampling was utilized to maintain the same class distribution in both sets, which is crucial for datasets with imbalanced classes.

### 2.5. Machine Learning Training

A systematic training process was implemented involving 24 machine learning classifiers drawn from five major algorithmic families, including ensemble methods, linear models, support vector machines, probabilistic models, and instance-based approaches. While we acknowledge that not all algorithms are equally suited to every dataset structure, this broad inclusion was intentionally designed to offer a representative and systematic comparison across commonly used ML approaches in clinical prediction tasks.

Each model was trained using default hyperparameters provided by the scikit-learn and XGBoost libraries. This choice was guided by the No Free Lunch Theorem, which states that no machine learning algorithm, or specific hyperparameter configuration, is universally optimal across all tasks or datasets [13]. Accordingly, we deliberately avoided tuning hyperparameters, opting for default values to ensure reproducibility, reduce the risk of overfitting, and maintain a fair comparison across models. While this approach may constrain peak performance, it provides a robust and standardized foundation for model evaluation.

To select the best-performing model, the Matthews Correlation Coefficient (MCC) was used as the primary evaluation metric on the test set. MCC is a robust metric that takes into account true and false positives and negatives and is particularly informative for imbalanced classification problems, providing a more balanced evaluation.

### 2.6. Algorithms Considered

All machine learning algorithms, reported in Appendix B, were evaluated, categorized based on their methodological approach.

### 2.7. Evaluation of the Best Model

The performance of the trained models was evaluated using two metrics: the Matthews Correlation Coefficient (MCC) and the Area Under the Receiver Operating Characteristic Curve (AUC-ROC) (Appendix A).

Informative in the context of class imbalance, the metrics account for different aspects of model performance. However, MCC was chosen as the primary criterion for model selection due to its robustness in imbalanced classification tasks. Unlike AUC, which focuses on discriminative capacity across thresholds, MCC reflects the overall balance between true and false classifications and is less affected by skewed class distributions. This allowed us to identify the model that maintained the best equilibrium between sensitivity and specificity across all outcome categories.

Moreover, we compared the chosen machine learning model with the logistic regression classifier, which represents a linear multivariate analysis. Receiver Operating Characteristic (ROC) curves were plotted to analyze the model’s performance across different threshold settings, highlighting the trade-offs between sensitivity (true positive rate) and specificity (false positive rate).

Model performance was evaluated using a single hold-out validation split (80% training, 20% test), chosen to maintain consistency across tasks and facilitate controlled comparisons. This design is focused on the exploration and comparison of the behavior of different machine learning algorithms across clinical prediction tasks, with a particular emphasis on explainability through XAI techniques.

While we acknowledge that this approach does not account for variance introduced by different data partitions, we addressed the robustness of the reported performance metrics by applying bootstrap resampling (1000 iterations) on the test set. This allowed us to compute 95% confidence intervals (CIs) for AUC and MCC.

### 2.8. Explainable Artificial Intelligence (XAI)

To improve model interpretability and better understand the decision-making process of the trained machine learning models, we employed SHAP (SHapley Additive exPlanations), a model-agnostic explainable AI (XAI) technique. SHAP values provide a consistent and theoretically grounded method for quantifying the contribution of each input feature to a specific prediction.

In our analysis, we generated SHAP summary plots to visualize the magnitude and direction of feature effects across the dataset. These plots allow us to identify which variables most consistently influenced the model’s output and how changes in feature values impacted the prediction risk. The objective of this analysis is to validate the plausibility of the model’s behavior by comparing it with established clinical knowledge. This form of validation is essential in clinical contexts, where models must not only be accurate but also transparent and aligned with domain expectations. Moreover, SHAP helped identify unexpected feature influences, providing opportunities to uncover novel associations that could inform future clinical hypotheses or decision support strategies.

## 3. Results

A total of 216 patients (47.7% male) were included in the study. A total of 122 of them (56.5%) developed POPF following pancreaticoduodenectomy, with 71 cases (32.8%) classified as Grade A and 51 (23.6%) as Grade B-C.

Patient demographics, preoperative data, surgical characteristics, and postoperative data are reported in Table 1 and Table 2. As summarized in Table 3 and Table 4, patients with POPF and those with no POPF had a similar age distribution (71 [66–75] vs. 70 [63–76] vs. 73 [66–77] years, *p* = 0.395). Operative time was significantly longer in patients who developed CR-POPF: a median of 358.5 min (IQR: 311–418) in those without a fistula, 360 min (IQR: 312–423) in Grade A cases, and 395 min (IQR: 347–466) in Grade B-C cases (*p* = 0.0169). In particular, there were no significant differences between the no fistula and Grade A groups (*p* = 0.9659), but significant differences were observed between the no fistula and Grade B-C groups (*p* = 0.0089), as well as between the Grade A and Grade B-C groups (*p* = 0.0122). In contrast, intraoperative blood loss did not show a significant impact on POPF development, with similar median values across groups (200 [170–390] mL vs. 260 [200–390] mL vs. 250 [200–390] mL, *p* = 0.9028). Similarly, BMI was not significantly associated with POPF occurrence, with median values of 23.4 (IQR: 21.1–26.8) in patients without fistula, 24.2 (IQR: 22.2–26.3) in Grade A cases, and 24.7 (IQR: 22.8–27.7) in Grade B-C cases (*p* = 0.395). Regarding several clinical and surgical factors, male gender was associated with a higher incidence of POPF (*p* = 0.026), along with hypertension (*p* = 0.042) and diabetes (*p* = 0.016). Neoadjuvant therapy (*p* = 0.031) correlates with a reduced risk of POPF. Among surgical factors, the type of PD significantly affected fistula risk (*p* = 0.042), with the Traverso–Longmire procedure associated with higher rates compared to Whipple. Vascular resection (*p* = 0.026) also correlates with increased POPF incidence. Additionally, two strong predictors of fistula formation were a soft pancreas (*p* = 0.033) and a Wirsung duct ≤ 3 mm (*p* < 0.001). In contrast, surgical approach (open vs. laparoscopic) did not impact fistula rates (*p* = 0.131), nor did histology positive for adenocarcinoma (*p* = 0.059), ASA classification (*p* = 0.421), smoking (*p* = 0.258), or bile culture (*p* = 0.161).

In addition, Table 2 and Table 4 summarize the detailed data on bile cultures and postoperative biochemical data. More specifically, the analysis showed that, among the individual bacterial species analyzed, positivity for *E. faecium* was found to be a statistically significant risk factor for POPF (*p* = 0.032). Additionally, postoperative data analysis revealed significant associations between right and left drainage amylase levels in POD III (*p* < 0.001); serum lipase levels in POD I, II, and III (*p* < 0.001); and PCT levels in POD III (*p* < 0.001) and the occurrence of POPF.

The performance of the best-performing machine learning models was assessed using the Matthews Correlation Coefficient (MCC) and the Area Under the Receiver Operating Characteristic Curve (AUC-ROC). Based on MCC, GradientBoostingClassifier was determined to be the best-performing model and achieved an AUC of 0.88 (95% CI: 0.78–0.96) and a Matthews Correlation Coefficient (MCC) of 0.68 (95% CI: 0.49–0.85).

To further evaluate and compare model performance across the no fistula group, Grade A fistula group and Grade B/C fistula group, AUC-ROC analysis identified multiple top-performing models. For the no fistula group, the XGBClassifier achieved the highest AUC of 0.99, followed closely by GradientBoostingClassifier at 0.98, and RandomForestClassifier with an AUC of 0.96. When tested on the Grade A fistula group, RandomForestClassifier led with an AUC of 0.88, followed closely by the XGBRFClassifier (AUC 0.86) and GradientBoostingClassifier (AUC 0.87). XGBClassifier also showed strong performance with an AUC of 0.83. For the Grade B-C fistula group, RandomForestClassifier (AUC 0.86) and XGBRFClassifier (AUC 0.85) were the best performers, with GradientBoostingClassifier achieving an AUC of 0.79 and XGBClassifier scoring 0.79 as well. Logistic regression was also included in the analysis, treated as another classification algorithm. Compared to the other models, it underperformed across all fistula groups, achieving an AUC of 0.84 for the no fistula group, 0.43 for the Grade A fistula group, and 0.58 for the Grade B-C fistula group. Additionally, SHAP (SHapley Additive exPlanations) was employed to analyze the impact of individual variables on model outputs, providing insights into which features were driving predictions. The performance of the best-performing models is summarized in terms of AUC and MCC. In addition, ROC curves were plotted to illustrate the diagnostic performance across different threshold levels (Figure 1), confirming the superior discriminative capacity of the GradientBoostingClassifier compared to traditional logistic regression. The SHAP analysis results, including summary plots, are shown in Figure 2.

## 4. Discussion

Improvements in healthcare quality, including surgical care, have been observed worldwide [14]. A similar trend has been observed specifically in pancreatic surgery, with substantial decreases and stabilization of mortality rates in high-volume centers within highly developed countries, ranging between 0% and 3% [15].

Focusing on proximal pancreatic resections, relevant to the present study population, the primary outcome data appear less encouraging. Recent estimates indicate a major morbidity rate of approximately 26% [16]. POPF remains the most frequent and stubborn complication despite substantial clinical efforts and the pancreatic community’s ongoing commitment to prevention. Indeed, a recent study of 2501 PDs estimated a global POPF incidence of 28%, with clinically relevant grades (B and C) at 19.7% [17]. Consequences of POPF notably include secondary complications such as intra-abdominal abscesses, sepsis, and potentially life-threatening hemorrhages, which further prolong hospital stays, elevate healthcare costs, and, notably, jeopardize the patients’ adjuvant therapy pathways [18,19].

The accurate identification of patients at high risk for POPF is currently a priority, yet significant challenges remain. The scientific community has actively engaged in developing predictive models to stratify POPF risk and optimize clinical resource allocation, focusing on patients who could benefit the most [20]. Current clinical practice relies on two fundamental pillars: intraoperative POPF risk assessment and an early drain removal or selective drain placement policy [21].

Intraoperative risk stratification utilizes scores based on static preoperative and intraoperative variables. Among these, the Fistula Risk Score (FRS) by Callery et al. is the most widely used. This system, based on a 10-point scale, combines four key risk factors: pancreatic gland texture, histology, pancreatic duct diameter, and intraoperative blood loss [6].

A modified version developed later by Kantor et al. integrates gender, BMI, total bilirubin, pancreatic duct diameter, and gland texture, achieving acceptable C-index values of 0.70 for test sets, 0.70 for internal validation, and 0.62 for external validation [22].

Timothy Mungroop et al. proposed a simplified model (alternative-FRS, a-FRS), based on BMI, pancreatic texture, and duct diameter, validating it across two independent databases. This model demonstrated superior predictive ability compared to the original FRS (AUC: 0.72 vs. 0.70, *p* = 0.05) and good discrimination (0.75 internal dataset, 0.78 external) [23].

Nevertheless, these models’ limitations were highlighted by a study by Rajesh S. Shinde et al., which compared their predictive accuracy in a cohort of 825 PD patients, showing moderate predictive capability across all scores analyzed. The results showed an AUC of 0.65 for FRS and 0.69 for a-FRS, with statistically significant differences between the models (*p* = 0.006) [13].

Thus, current predictive models for POPF have significant limitations, with suboptimal discriminative capabilities, hindering effective risk stratification and prevention strategies. This scenario highlights the need for more advanced tools capable of overcoming traditional models’ methodological limitations, thereby improving postoperative clinical management. Traditional models, including FRS and variants, rely on static pre- and intraoperative factors, neglecting the dynamic clinical evolution of complications like POPF, reducing their reliability and clinical applicability.

ML presents a concrete opportunity to refine POPF risk stratification, given its ability to analyze complex clinical variable relationships dynamically [24]. Integrating postoperative data into predictive models could be a turning point, allowing the timely identification of at-risk patients, optimizing perioperative management, and personalizing therapeutic strategies.

In recent years, over ten ML-based POPF prediction models have been developed [25,26]. Although ML models are often perceived as superior to traditional regression models, a recent study show no superiority of ML over logistic regression for predicting POPF after PD [27].

Our study investigated ML applications in predicting POPF after PD, analyzing a large single-center cohort operated by an expert surgeon. Specifically, it evaluated correlations among clinical parameters (pre-, intra-, and early postoperative until POD 3) and POPF-related events (absence, BL), and CR-POPF. Twenty-four ML algorithms were tested to identify the best-performing model, with GradientBoostingClassifier showing the highest predictive accuracy. Routine clinical variables contributed substantially to model performance, as demonstrated by consistently high values of AUC and MCC, metrics chosen specifically for their robustness in imbalanced classification settings. Notably, influential risk factors identified included drain fluid amylase (DFA) on POD 3, serum lipase on POD 1 and POD 3, POD 3 PCT, and operation time.

The detailed SHAP value analysis highlighted DFA on the left drain as particularly influential in predicting POPF absence, followed by other variables such as right-drain DFA, POD 3 lipase, and PCT. The influence of DFA was less dominant for predicting CR-POPF, where PCT became comparably influential. PCT has been indeed recently validated as a significant biomarker for predicting CR-POPF. Coppola et al. demonstrated that low postoperative PCT levels are highly effective in excluding the onset of CR-POPF following PD, highlighting its utility as a reliable early postoperative marker [28].

The SHAP analysis (Figure 1) provided a model-agnostic interpretation of feature relevance, confirming that early postoperative biochemical variables—particularly drain fluid amylase from the left drain, serum lipase, and procalcitonin—were the most influential contributors to the model’s predictions. Interestingly, DFA was most associated with the prediction of POPF absence, whereas PCT gained greater relevance in the prediction of CR-POPF. These patterns not only mirror clinical experience but also support the internal validity of the model and its potential to generate physiologically consistent outputs. The integration of explainable AI tools, such as SHAP, contributes to the model’s transparency and may facilitate clinical acceptance and implementation.

Our model, especially in predicting the absence of POPF, highlighted a significant influence of serum lipase on the POD3. Specifically, the SHAP feature importance plot underlines a significant relationship between low lipase values and the absence of POPF. In 2022, the ISGPS clearly defined the entity of post-pancreatectomy acute pancreatitis (PPAP), assigning a pivotal diagnostic role to postoperative hyperamylasemia [29]. However, postoperative hyperlipasemia is emerging as predictive for POPF. Tang et al. (2025) showed POD 1 lipase > 60 U/L independently predicted POPF, with a superior accuracy (AUC 0.791) compared to serum amylase [30].

Our ML model significantly outperformed logistic regression across POPF outcomes (absence: AUC 0.98 vs. 0.84; BL: AUC 0.87 vs. 0.43; CR-POPF: AUC 0.79 vs. 0.58), emphasizing the value of including early postoperative variables.

These findings support the potential integration of the model into postoperative workflows to guide risk-adapted clinical decisions. As a future perspective, the model could serve as the basis for a standardized protocol for drain management after pancreaticoduodenectomy.

Specifically, patients identified as low-risk for CR-POPF on postoperative day 3 may be considered for early drain removal, in the absence of clinical signs of complication, with the aim of reducing drain-related morbidity and accelerating recovery. Conversely, intermediate- and high-risk patients would continue under conventional management, with delayed removal guided by clinical assessment and postoperative biomarkers such as DFA, lipase, and procalcitonin.

The impact of such a strategy could be retrospectively assessed within our current cohort, using clinical endpoints including the rate of secondary interventions, length of hospital stay, and 30-day readmissions. This risk-based approach may represent a concrete step toward dynamic and personalized postoperative care supported by a machine learning tool. Our study undoubtedly has some limitations. First, the retrospective nature of the analysis inevitably introduces selection bias, as all data come from a single center.

Another important limitation of our work is the absence of external validation. Although the use of a single-center dataset ensured consistency in surgical technique, perioperative management, and data collection, it inevitably limits the generalizability of the findings.

To enhance reproducibility and reduce center-specific bias, the model was built using only objective, routinely available variables, avoiding subjective assessments. However, differences in local protocols—such as timing of biomarker monitoring or drain management strategies—may still influence its performance in other settings and should be considered in future validations.

Future efforts should therefore focus on validating the model in external cohorts from other high-volume pancreatic surgery centers to confirm its robustness across varying institutional practices and patient populations. Finally, although our ML algorithms can evaluate the risk of POPF, the large number of variables included in the model could limit its clinical applicability.

## 5. Conclusions

Our study demonstrates how the application of ML models allows for high accuracy in predicting the risk of developing POPF after PD. The use of ML permits synergistic processing of a wide set of clinical, surgical, and early postoperative variables, producing a robust and high-performing predictive model.

The developed model proved capable of precisely identifying, within the third postoperative day, patients at low risk of POPF who can safely undergo more selective and personalized postoperative management strategies, such as early surgical drain removal, thus enabling a more favorable postoperative course and simultaneously allocating clinical resources toward higher-risk patients.

In addition to its predictive accuracy, the proposed model may serve as a foundation for future risk-adapted management strategies, particularly in guiding early postoperative decisions such as drain removal. Its integration into clinical practice could facilitate more personalized and dynamic postoperative care, ultimately improving outcomes and optimizing the use of healthcare resources.

However, prospective and multicenter studies will be necessary to further validate the model and define its impact on clinical management and long-term outcomes.

Finally, the integration of explainable AI methods, such as SHAP, enhances interpretability and may facilitate the clinical adoption of predictive models by offering physiologically meaningful, transparent decision support.

## Figures and Tables

**Figure 1 cancers-17-01846-f001:**
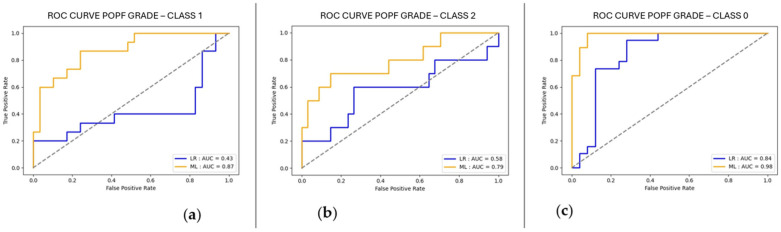
ROC curves comparing the performance of logistic regression (LR) and the selected machine learning (ML) model in predicting postoperative pancreatic fistula (POPF) grades: (**a**) Class 1 (biochemical leak), (**b**) Class 2 (clinically relevant fistula), and (**c**) Class 0 (absence of fistula). The ML model consistently outperforms logistic regression across all classes, with AUC values of 0.87, 0.79, and 0.98 for Classes 1, 2, and 0, respectively.

**Figure 2 cancers-17-01846-f002:**
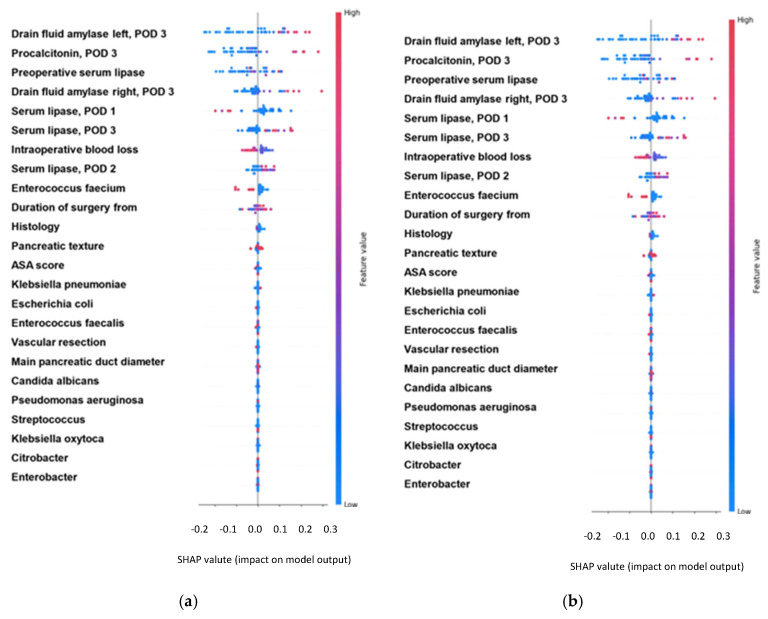
SHAP analysis summary plots, showing the impact of variables on the ML model outputs. (**a**) The most relevant features to predict the absence of POPF after the PD procedure by the GradientBoostingClassifier algorithm using the dataset. (**b**) The most relevant features to predict CR-POPF after the PD procedure by the GradientBoostingClassifier algorithm using the dataset.

**Table 1 cancers-17-01846-t001:** Patient characteristics.

Variable	N = 216
Age (median [IQR]), years	71 [65–76]
Sex (%)	
Male	103 (47.7)
Female	113 (52.3)
BMI (median [IQR]), kg/m^2^	23.94 [21.30–26.83]
Cardiovascular comorbidity (%)	67 (31)
Pulmonary comorbidity (%)	30 (13.9)
Diabetes (%)	38 (17.6)
Smoker (%)	45 (20.8)
Hypertension (%)	104 (48.1)
ASA classification (%)	
1	9 (4.2)
2	97 (44.9)
3–4	110 (51)
Neoadjuvant therapy (%)	25 (11.5)
Positive bile culture (%)	128 (59.3)
Surgical approach: Open (%)	206 (95.4)
Whipple (%)	75 (34.7)
Traverso (%)	141 (65.3)
Texture (%)	
Soft	130 (60.19)
Firm	86 (39.81)
Diameter of MPD (%)	
<3 mm	85 (39.4)
>3 mm	131 (60.6)
Surgery time (median [IQR]), min	363.00 [321.00, 420.00]
Intraoperative bool loss (median [IQR]), mL	260.00 [170.00, 390.00]
Vascular resection (%)	42 (19.4)
Pathology: PDAC (%)	170 (78.7)

IQR: interquartile range; BMI: Body mass index; ASA: American Society of Anesthesiologists; MPD: Main pancreatic duct; PDAC: Pancreatic ductal adenocarcinoma.

**Table 2 cancers-17-01846-t002:** Postoperative data.

Variable	N = 216
Amylase right-drain PODIII (median [IQR]), UI/L	156 [23–676.5]
Amylase left-drain PODIII (median [IQR]), UI/L	227.5 [23.5–1223]
Blood lipase PODI (median [IQR]), UI/L	101.95 [30.6–403.75]
Blood lipase PODII (median [IQR]), UI/L	52 [22.25–193.05]
Blood lipase PODII (median [IQR]), UI/L	32.6 [20.61–72.55]
PCT PODIII (median [IQR]), UI/L	0.375 [0.195–0.845]
Klebsiella pneumoniae (%)	47 (21.8)
Klebsiella oxytoca (%)	19 (8.8)
Enterobacter (%)	22 (10.2)
Pseudomonas aeuriginosa (%)	12 (5.5)
Citrobacter (%)	18 (8.3)
Enterococcus faecalis (%)	62 (28.7)
Enterococcus faecium (%)	58 (26.9)
Streptococcus (%)	23 (10.6)
Candida albicans (%)	14 (6.5)

IQR: interquartile range; POD: postoperative date.

**Table 3 cancers-17-01846-t003:** Risk factors and POPF.

Variable	No Fistula (n = 94)	Grade A (n = 71)	Grade B-C (n = 51)	*p* Value
Sex				0.026
Male	35 (37.2%)	39 (54.9%)	29 (56.9%)	
Female	59 (62.8%)	32 (45.1%)	22 (43.1%)	
Cardiovascular Diseases				0.125
No	63 (67%)	55 (77.5%)	31 (60.8%)	
Yes	31 (33%)	16 (22.5%)	20 (39.2%)	
Respiratory Diseases				0.456
No	78 (83%)	62 (87.3%)	46 (90.2%)	
Yes	16 (17%)	9 (12.7%)	5 (9.8%)	
Smoking				0.258
Non-smoker	63 (67%)	53 (74.6%)	31 (60.8%)	
Smoker	31 (33%)	18 (25.4%)	20 (39.2%)	
Hypertension				0.042
No	41 (43.6%)	45 (63.4%)	26 (51%)	
Yes	53 (56.4%)	26 (36.6%)	25 (49%)	
Diabetes				0.016
No	70 (74.5%)	65 (91.5%)	43 (84.3%)	
Yes	24 (25.5%)	6 (8.5%)	8 (15.7%)	
Histology				0.059
PDAC	81 (86.2%)	51 (71.8%)	38 (74.5%)	
Other Types	13 (13.8%)	20 (28.2%)	13 (25.5%)	
Neoadjuvant Treatment				0.031
No	77 (81.9%)	66 (93.7%)	48 (94.1%)	
Yes	17 (18.1%)	5 (6.3%)	3 (5.9%)	
ASA Classification				0.421
ASA 1	5 (5.3%)	1 (1.4%)	3 (5.9%)	
ASA 2	41 (43.6%)	37 (52.1%)	19 (37.3%)	
ASA 3	45 (47.9%)	28 (39.4%)	25 (49%)	
ASA 4	3 (3.2%)	5 (7%)	4 (7.8%)	
MPD Diameter				<0.001
<3 mm	21 (22.3%)	33 (46.5%)	31 (60.8%)	
≥3 mm	73 (77.7%)	38 (53.5%)	20 (39.2%)	
Vascular Resection				0.026
No	68 (72.3%)	61 (85.9%)	45 (88.2%)	
Yes	26 (27.7%)	10 (14.1%)	6 (11.8%)	
Pancreatic Texture				0.033
Soft	46 (49%)	26 (36.6%)	14 (27.5%)	
Firm	48 (51%)	45 (63.4%)	37 (72.5%)	
Surgical Approach				0.131
Open	91 (96.8%)	69 (97.2%)	46 (90.2%)	
Minimally Invasive	3 (3.2%)	2 (2.8%)	5 (9.8%)	
Whipple	35 (37.2%)	17 (23.9%)	23 (45.1%)	0.042
Traverso	59 (62.8%)	54 (76.1%)	28 (54.9%)	
Bile Culture				0.161
Positive	45 (47.9%)	24 (33.8%)	19 (37.3%)	
Negative	49 (52.1%)	47 (66.2%)	32 (62.7%)	
Age (Median [IQR], years)	70 [63–76]	71 [66–75]	71 [66–75]	0.395
BMI (Median [IQR], kg/m²)	23.4 [21.1–26.8]	24.2 [22.2–26.3]	24.7 [22.8–27.7]	0.395
Surgical Time (Median [IQR], min)	358.5 [311–418]	360 [312–423]	395 [347–466]	0.0169
Intraoperative Blood Loss (Median [IQR], mL)	200 [170–390]	260 [200–390]	250 [200–390]	0.9028

IQR: Interquartile range; PDAC: Pancreatic ductal adenocarcinoma; ASA: American Society of Anesthesiologists; MPD: Main pancreatic duct; BMI: Body mass index.

**Table 4 cancers-17-01846-t004:** Postoperative risk factors and POPF.

Variable	No Fistula (n = 94)	Grade A (n = 71)	Grade B-C (n = 51)	*p* Value
Klebsiella Pneumoniae				0.633
No	76 (80.9%)	53 (74.6%)	40 (78.4%)	
Yes	18 (19.1%)	18 (25.4%)	11 (21.6%)	
Klebsiella Oxytoca				0.676
No	87 (92.6%)	65 (91.5%)	45 (88.2%)	
Yes	7 (7.4%)	6 (8.5%)	6 (11.8%)	
Enterobacter				0.476
No	84 (89.4%)	66 (93%)	44 (86.3%)	
Yes	10 (10.6%)	5 (7%)	7 (13.7%)	
Pseudomonas Aeuriginosa				0.154
No	92 (97.9%)	65 (91.5%)	47 (92.2%)	
Yes	2 (2.1%)	6 (8.5%)	4 7.8%)	
Citrobacter				0.886
No	87 (92.6%)	65 (91.5%)	46 (90.2%)	
Yes	7 (7.4%)	6 (8.5%)	5 (9.8%)	
Enterococcus Faecalis				0.600
No	70 (74.5%)	50 (70.4%)	34 (66.7%)	
Yes	24 (25.5%)	21 (29.6%)	17 (33.3%)	
Enterococcus Faecium				0.032
No	75 (79.8%)	44 (62.0%)	39 (76.5%)	
Yes	19 (20.2%)	27 (38%)	12 (23.5%)	
Streptococcus				0.205
No	80 (85.1%)	66 (93%)	47 (92.2%)	
Yes	14 (14.9%)	5 (7%)	4 (7.8%)	
Candida Albicans				0.529
No	86 (91.5%)	67 (94.4%)	49 (96.1%)	
Yes	8 (8.5%)	4 (5.6%)	2 (3.9%)	
Amylase right-drain PODIII (median [IQR]), UI/L	17.5 [11–45]	402 [160–964]	877 [222–1940]	<0.001
Amylase left-drain PODIII (median [IQR]), UI/L	18.5 [9–82]	585 [222–1652]	1758 [336–5580]	<0.001
Blood lipase PODI (median [IQR]), UI/L	30 [17.5–88]	329 [68.9–1004]	237.2 [104.7–618]	<0.001
Blood lipase PODII (median [IQR]), UI/L	22 [16.4–44.97]	157 [45.9–404.9]	164 [50.2–409.8]	<0.001
Blood lipase PODIII (median [IQR]), UI/L	21.65 [15.1–34]	61.5 [30–131]	46.93 [23.7–226]	<0.001
PCT PODIII (median [IQR]), UI/L	0.23 [0.13–0.52]	0.36 [0.22–0.73]	0.97 [0.46–3.2]	<0.001

IQR: Interquartile range; POD: Postoperative date.

## Data Availability

The analysis code and the data adopted and discussed in this study are available upon request from the corresponding author.

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
