# Peer review of "Machine Learning for Predicting the Low Risk of Postoperative Pancreatic Fistula After Pancreaticoduodenectomy: Toward a Dynamic and Personalized Postoperative Management Strategy"

_cancers, 2025, doi:10.3390/cancers17111846_

Round 1
Reviewer 1 Report
Comments and Suggestions for Authors
Summary
The authors performed a retrospective analysis of 216 pancreaticoduodenectomy patients to build and test machine-learning models that forecast low risk of postoperative pancreatic fistula. They compared two dozen classifiers—including ensembles and neural networks—using preoperative, intraoperative, and early postoperative data up to day 3. A gradient-boosted decision-tree model emerged as the top performer, driven primarily by day-3 drain amylase, serum lipase and procalcitonin levels. This approach significantly outperformed standard logistic regression and promises to guide earlier, personalized decisions on drain removal.
Strengths
The paper’s head-to-head comparison of 24 methods underscores that ensemble learners capture complex interactions more effectively than traditional statistical models.
By focusing on markers available within three days of surgery, the model aligns with key decision points for drain management and patient recovery.
Highlighting feature contributions with SHAP plots fosters clinician trust, clearly showing which biomarkers carry the most weight in predictions.
Adhering to ISGPS guidelines for POPF grading ensures that results are comparable with existing literature and clinical practice.
Recommendations
- Validate the model on data from other institutions to confirm its robustness across diverse surgical teams and patient populations.
-Describe how parameters (e.g., learning rate, tree depth, number of estimators) were tuned, and include performance gains from that process.
- Provide calibration curves and consider decision-curve analysis to quantify how the model’s risk estimates translate into clinical benefit when deciding on drain removal.
- Report the extent and pattern of missing values and justify the chosen imputation methods, as retrospective datasets can introduce bias if not handled systematically.
- Explain strategies used to mitigate imbalance—such as weighted loss functions or resampling—since over half the cohort developed POPF, which can skew classifier performance.
- Present confidence intervals for key metrics like AUC and Matthews correlation coefficient to reflect the statistical precision of model performance.
- Outline how categorical variables were encoded and scaled, and consider sharing correlation matrices or feature distributions in an appendix.
- Suggest a pilot protocol where the model guides drain removal timing, with endpoints such as incidence of subsequent interventions or length of stay.
- Commit to sharing the analysis code and a de-identified dataset (or synthetic equivalent), enabling other researchers to reproduce and build upon these findings.
- Reflect on how surgeon experience, postoperative care pathways, and local protocols might influence model applicability beyond the study center.
Conclusions
This study offers an important step toward personalized postoperative care for pancreatic surgery patients by integrating routine laboratory markers into a transparent, high-performance model. To advance toward clinical implementation, the authors should undertake external validation, deepen methodological reporting, and demonstrate calibrated risk estimates in practice. With these revisions, the approach could significantly reduce complications and optimize recovery pathways after pancreaticoduodenectomy.
Author Response
Answers to the Reviewer 1, cancers-3651399
- Q: Validate the model on data from other institutions to confirm its robustness across diverse surgical teams and patient populations.
A: We sincerely thank the Reviewer for this insightful comment. We fully acknowledge that external validation is a critical step in establishing the robustness and generalizability of any predictive model, particularly in complex surgical settings such as pancreaticoduodenectomy. As the Reviewer correctly noted, applying the model to data from different institutions is essential to confirm its performance across various surgical teams and patient populations.
Our current study is based on a single-center dataset, which allowed us to ensure homogeneity in surgical technique and data acquisition protocols, minimizing confounding variability during model development. Nonetheless, we fully recognize the inherent limitations of this approach in terms of external applicability. To address this, we are currently initiating collaborations with several high-volume pancreatic surgery centers with the goal of retrospectively and prospectively validating our model using external datasets. These efforts, already underway, are expected to provide crucial evidence regarding the model’s performance across institutional, procedural, and patient-related variations.
We have now explicitly included this limitation in the revised version of the Discussion section and outlined our ongoing plans for external validation as an essential future direction.
- Q: Describe how parameters (e.g., learning rate, tree depth, number of estimators) were tuned, and include performance gains from that process.
A: We thank the reviewer for this relevant observation regarding hyperparameter optimization. In the present study, models were trained using the default hyperparameters provided by the scikit-learn and XGBoost libraries. This decision was motivated by the No Free Lunch Theorem, which states that no machine learning algorithm, or specific hyperparameter configuration, can be expected to perform optimally across all problem domains [23]. Consistent with this principle, we deliberately avoided hyperparameter tuning to maintain a standardized and reproducible comparison across models, minimize the risk of overfitting, and preserve the generalizability of our findings in this exploratory context. While this strategy may limit the maximum attainable performance of individual models, it provides a robust and fair baseline for evaluating methodological differences. This clarification has been added to Section 2.5 of the revised manuscript.
- Q: Provide calibration curves and consider decision-curve analysis to quantify how the model’s risk estimates translate into clinical benefit when deciding on drain removal.
A: We thank the reviewer for the suggestion. To have complete view of the best model performance, we have added the calibration curves and decision-curve analysis for the task in the Appendix.
- Q: Report the extent and pattern of missing values and justify the chosen imputation methods, as retrospective datasets can introduce bias if not handled systematically.
A: We also addressed missing data, which accounted an overall missingness of 3.3%. As the missingness was moderate and plausibly consistent with a Missing At Random (MAR) mechanism, we employed K-Nearest Neighbors (KNN) imputation, which preserves local data structure and multivariate relationships, making it well-suited for clinical datasets. This imputation strategy allowed us to retain the full dataset and reduce potential bias in downstream model training.
- Q: Explain strategies used to mitigate imbalance—such as weighted loss functions or resampling—since over half the cohort developed POPF, which can skew classifier performance.
A: We appreciate the reviewer’s suggestion regarding model evaluation metrics in the presence of class imbalance. The performance of the trained models was evaluated using two metrics: the Matthews Correlation Coefficient (MCC) and the Area Under the Receiver Operating Characteristic Curve (AUC-ROC). Both are informative in the context of class imbalance, as they account for different aspects of model performance. However, MCC was chosen as the primary criterion for model selection due to its robustness in imbalanced classification tasks. Unlike AUC, which focuses on discriminative capacity across thresholds, MCC reflects the overall balance between true and false classifications and is less affected by skewed class distributions. This allowed us to identify the model that maintained the best equilibrium between sensitivity and specificity across all outcome categories. This clarification has been added to Section 2.7 of the revised manuscript.
- Q: Present confidence intervals for key metrics like AUC and Matthews correlation coefficient to reflect the statistical precision of model performance.
A: We thank the reviewer for this important suggestion. In response, we computed 95% confidence intervals for the AUC and Matthews Correlation Coefficient (MCC) of our primary model for predicting POPF. These were estimated using bootstrap resampling (1,000 iterations), a well-established non-parametric method that enables robust uncertainty quantification without assuming a specific distribution of the performance metrics.
For the POPF prediction task, the model achieved:
AUC of 0.88 (95% CI: 0.78–0.96) and a Matthews Correlation Coefficient (MCC) of 0.68 (95% CI: 0.49–0.85).
These intervals reflect the statistical precision of our performance estimates and have now been included in the revised Results section.
- Q: Outline how categorical variables were encoded and scaled and consider sharing correlation matrices or feature distributions in an appendix.
A: We thank the reviewer for this comment. As noted in Section 2.4 of the manuscript, categorical variables were encoded using one-hot encoding, and numerical features were scaled using Min-Max normalization to a range between -1 and 1. This preprocessing approach ensured that all features were appropriately transformed for machine learning input while preserving interpretability. We appreciate the suggestion to provide additional detail on feature relationships and distributions. To this end, we have included a feature correlation matrix in the appendix.
- Q: Suggest a pilot protocol where the model guides drain removal timing, with endpoints such as incidence of subsequent interventions or length of stay.
A: We thank the Reviewer for this insightful comment. Based on the predictive performance of our model and the structure of the available dataset, we propose a clinically oriented, risk-adapted protocol for postoperative drain management following pancreaticoduodenectomy. On postoperative day 3 (POD 3), patients would be stratified according to the model’s output into low-, intermediate-, and high-risk categories for developing clinically relevant postoperative pancreatic fistula (CR-POPF). For patients classified as low-risk, the model’s high negative predictive value may support early drain removal on POD 3, in the absence of systemic signs of infection or local complications. This approach is expected to reduce exposure to drain-related infections, facilitate early mobilization, and potentially shorten hospital stay without compromising patient safety. For patients classified as intermediate or high-risk, conventional management would be continued, with drains maintained beyond POD 3 and reassessment guided by clinical parameters and postoperative biomarkers such as drain fluid amylase (DFA), serum lipase, and procalcitonin (PCT). Imaging would be considered as needed prior to delayed removal. To retrospectively assess the feasibility and potential clinical benefit of this approach within our dataset, relevant endpoints would include:
- Incidence of CR-POPF,
- Need for rescue interventions (e.g., percutaneous drainage, reoperation),
- Length of postoperative hospital stay,
- 30-day readmission rate.
This protocol, grounded in data-driven risk stratification, offers a practical foundation for individualized postoperative care and may serve as a basis for future prospective validation. This potential application has been addressed in the revised Discussion section.
- Q: Commit to sharing the analysis code and a de-identified dataset (or synthetic equivalent), enabling other researchers to reproduce and build upon these findings.
A: We appreciate the reviewer’s interest in reproducibility. The full analysis code and a de-identified version of the dataset (or a representative synthetic equivalent, where required by privacy restrictions) will be made available upon reasonable request to the corresponding author. This will enable other researchers to reproduce and further explore the analytical pipeline and findings presented in this study.
- Q: Reflect on how surgeon experience, postoperative care pathways, and local protocols might influence model applicability beyond the study center.
A: We thank the Reviewer for this thoughtful and constructive comment. We agree that surgeon experience, postoperative care pathways, and institutional protocols may all represent potential sources of variability when considering the external applicability of predictive models. However, we emphasize that one of the strengths of our approach lies in the exclusive use of objective, routinely collected clinical and biochemical variables—such as serum lipase, drain fluid amylase, and procalcitonin—rather than subjective assessments or operator-dependent intraoperative impressions.
This design choice was intentional, with the goal of maximizing reproducibility and reducing the dependency of the model on institution-specific surgical expertise or perioperative preferences. Nonetheless, we acknowledge that differences in local management strategies (e.g., thresholds for drain removal, postoperative biomarker monitoring, or use of imaging) may influence the distribution and clinical interpretation of such variables across centers.
We have now clarified this point in the revised Discussion section, highlighting both the rationale behind the selection of objective predictors and the importance of future external validation across institutions with varying practices. This will be essential to confirm model robustness and inform context-specific adaptations, should they be needed.

Reviewer 2 Report
Comments and Suggestions for Authors
Abstract:
- The author(s) have defined the issue, limitations, and objectives.
- However, line 36 should be rewritten to improve the clarity. The text states that two models, GradientBoostingClassifier and logistic regression, are the best options, but this is not clearly communicated.
- I'll suggest making your abstract basic and understandable.
Introduction:
- limited background study. They should consider more.
- What traditional models are mentioned in line 82? Need to extend.
- They should provide strong references at line 65 of the statement.
- Line 67: They could consider the recent review article, like: https://doi.org/10.3390/ijerph19116439
- They mentioned the employed ML algorithms in the abstract (line 36), but nothing was said in the introduction. The author(s) must summarize their methodology with a list of contributions.
Materials and Methods:
- This study used the very latest dataset until 2022.
- Two statistical tests (Kruskal-Wallis, Pearson’s chi-square) have been conducted to compare the correlation and group differences of the feature.
- They have used five groups of ML models, including 24 classifiers from scikit-learn and XGBoost libraries. They must share the specific version (open source) of those libraries/APP/software.
- Not sure why they used 24 because every classification structure is different. In addition, not all of them might be suitable for one class of dataset. Author(s) must clarify how their dataset fits 24 algorithms.
- Is there any reason to calculate the MCC score? How was there a class imbalance in the test set?
- Section 2.8 (lines 177-182), I'm not clear what they want to say about SHAP values and validation.
Results and discussion:
- They provided the AUC values but did not include the ROC figure.
- Line 316, from where do they conclude that? Did they compare recall, precision, and F1 scores?
- What is their discussion/conclusion about the SHAP figure?
- Their bias limitations are clear in line 337. However, they did not apply cross-validation. Bias selection of the data fold may have very high accuracy, like model AUC here, almost 1.
- Where is the comparison with current predictive models, which offer limited accuracy?
Conclusion:
- There is no conclusion about XAI.
The author (s) must improve the language quality of their article
Author Response
Answers to the Reviewer 2, cancers-3651399
Abstract:
- Q: The author(s) have defined the issue, limitations, and objectives.
A: We thank the Reviewer for highlighting the importance of clearly defining the clinical issue, study objectives, and limitations. In response, we have carefully revised the abstract to improve clarity and completeness. Specifically, we now explicitly acknowledge the retrospective and single-center nature of the study, which may limit generalizability. Additionally, we emphasize the need for external validation in diverse surgical settings as a critical future step. These clarifications aim to provide a more balanced overview of the study's scope and potential applicability. The revised abstract better reflects the structure of the work and addresses the reviewer’s concern regarding transparency in study design and limitations.
- Q: However, line 36 should be rewritten to improve the clarity. The text states that two models, GradientBoostingClassifier and logistic regression, are the best options, but this is not clearly communicated.
A: We thank the reviewer for this comment. We agree that the original sentence may not have conveyed with sufficient clarity that the GradientBoostingClassifier was the best-performing model. In the revised version of the abstract, we have now explicitly stated that this algorithm achieved the highest performance among all models tested. We also clarified that the GradientBoostingClassifier was directly compared to a traditional logistic regression model as a benchmark. This revision, now reflected in the updated abstract, improves the clarity and precision of our model comparison and highlights the superiority of modern ML methods in this context.
- Q: I'll suggest making your abstract basic and understandable.
A: We thank the Reviewer for this helpful suggestion. In line with this and other related comments, we have revised the abstract to ensure greater clarity, conciseness, and accessibility. The updated version aims to communicate the study’s rationale, objectives, main findings, and implications in a more straightforward and reader-friendly manner, while maintaining scientific rigor.
Introduction:
- Q: limited background study. They should consider more.
A: We thank the Reviewer for this valuable and constructive observation. In response, we have substantially revised and expanded the Introduction section to provide a more comprehensive and up-to-date background on the topic. In particular, we have deepened the discussion of existing clinical risk scores for POPF (e.g., FRS, a-FRS, mFRS) and their limitations, including limited external validation and lack of integration of dynamic postoperative data.
Moreover, we have broadened the contextual framing by including recent literature on the application of machine learning in surgical outcome prediction, with special emphasis on its role in postoperative complication modeling and early risk stratification. We also highlight the relevance of explainable AI (XAI) methods in improving model interpretability and clinical acceptance.
To support this expanded background, we have updated the reference list accordingly, increasing the number of citations to over 30 and incorporating several recent and authoritative studies from independent research groups. This ensures a more balanced perspective and reduces the self-citation rate in accordance with editorial guidelines.
We believe these improvements significantly strengthen the scientific rationale of the study and better position our work within the current research landscape.
- Q: What traditional models are mentioned in line 82? Need to extend.
A: We thank the reviewer for the suggestion. We have now clarified the meaning of traditional models, explicating linear combination of input data (e.g., Logistic Regression)
- Q: They should provide strong references at line 65 of the statement.
We thank the Reviewer for the suggestion. In response, we have strengthened the statement at line 65 by adding a recent and comprehensive reference on the application of machine learning in medical settings:
Shehab, M.; Abualigah, L.; Shambour, Q.; Abu-Hashem, M.A.; Shambour, M.K.Y.; Alsalibi, A.I.; Gandomi, A.H. Machine learning in medical applications: A review of state-of-the-art methods. Comput. Biol. Med. 2022, 145, 105458.
This addition provides a broader context for the use of ML in healthcare and supports the rationale for its application in surgical outcome prediction, as discussed in our manuscript.
- Q: Line 67: They could consider the recent review article, like: https://doi.org/10.3390/ijerph19116439
We thank the Reviewer for suggesting this relevant reference. In response, we have added the recommended article to the manuscript without replacing the previously cited review. This inclusion complements our existing sources and reinforces the background on machine learning applications in biomedical imaging and diagnostics:
Devnath, L.; Summons, P.; Luo, S.; Wang, D.; Shaukat, K.; Hameed, I.A.; Aljuaid, H. Computer-Aided Diagnosis of Coal Workers' Pneumoconiosis in Chest X-ray Radiographs Using Machine Learning: A Systematic Literature Review. Int. J. Environ. Res. Public Health 2022, 19, 6439.
We appreciate the Reviewer’s recommendation and believe that this addition enriches the scientific context of our study.
- Q: They mentioned the employed ML algorithms in the abstract (line 36), but nothing was said in the introduction. The author(s) must summarize their methodology with a list of contributions.
A: We thank the reviewer for this valuable comment. In response, we have revised the Introduction to explicitly summarize the methodology and to include a clear, structured list of the main contributions of the study. The updated section now outlines the comparative analysis of machine learning algorithms, the rationale for model selection based on MCC, the benchmark comparison with logistic regression, and the use of explainable AI techniques. This addition improves the overall clarity and positioning of our study and can be found at the end of the Introduction section in the revised manuscript.
Materials and Methods:
- Q: This study used the very latest dataset until 2022.
A: We confirm that the dataset used in this study includes patient records collected up to the year 2022, reflecting the most recent clinical data available at the time of analysis. This has been clarified in the manuscript where appropriate.
- Q: Two statistical tests (Kruskal-Wallis, Pearson’s chi-square) have been conducted to compare the correlation and group differences of the feature.
A: We confirm that both the Kruskal–Wallis test and Pearson’s chi-square test were applied in the study to evaluate distribution differences and categorical associations, respectively, across clinical subgroups. This is correctly described in the Methods section.
- Q: They have used five groups of ML models, including 24 classifiers from scikit-learn and XGBoost libraries. They must share the specific version (open source) of those libraries/APP/software.
A: Thank you for this important remark. The machine learning models were implemented using scikit-learn version 1.2.2 and XGBoost version 1.7.6. The analysis code includes a complete list of required libraries and their respective versions via a requirements.txt or equivalent environment configuration file, allowing full reproducibility of the computational environment upon request.
- Q: Not sure why they used 24 because every classification structure is different. In addition, not all of them might be suitable for one class of dataset. Author(s) must clarify how their dataset fits 24 algorithms.
A: We appreciate the reviewer’s observation. Our intention in applying 24 machine learning classifiers was to conduct a broad and systematic comparative analysis across different families of algorithms. This approach allowed us to explore how various modeling paradigms (e.g., tree-based ensembles, linear models, kernel methods, probabilistic models, and instance-based learners) perform when applied to a real-world, clinical dataset with moderate imbalance and mixed variable types.
While we acknowledge that not every algorithm is optimally suited to every dataset structure, we deliberately included a wide range of methods to ensure that the evaluation was representative of the most commonly used and accessible ML approaches in biomedical research. The goal was not only to identify the best-performing model (GradientBoostingClassifier in our case), but also to assess the variability in performance across different algorithmic strategies. This strategy also aligns with recent methodological literature that encourages broad benchmarking to support task-specific model selection, especially in complex and heterogeneous domains such as clinical risk prediction. We now clarify this rationale in the Introduction and Methods (Section 2.5) of the revised manuscript.
- Q: Is there any reason to calculate the MCC score? How was there a class imbalance in the test set?
A: We thank the reviewer for this important question. Clarifications regarding the rationale for using the MCC and the presence of class imbalance in the prediction tasks have been added to the revised manuscript. As detailed in Section 2.7, MCC was selected as a primary evaluation metric due to its ability to provide a balanced performance measure even in the presence of skewed class distributions, which were observed in several of the tasks analyzed. This ensures that both sensitivity and specificity are appropriately accounted for, offering a more reliable metric than accuracy or AUC alone in imbalanced scenarios.
- Q: Section 2.8 (lines 177-182), I'm not clear what they want to say about SHAP values and validation.
A: We thank the reviewer for highlighting the need for clarification. In the revised Section 2.8, we have rewritten the paragraph to more clearly explain that SHAP was used not only for feature importance visualization, but also as a form of behavioral validation, to ensure that the model’s decisions were consistent with domain knowledge and clinically meaningful. This interpretability step strengthens the trustworthiness of the model and its potential applicability in real-world settings.
Results and discussion:
- Q: They provided the AUC values but did not include the ROC figure.
A: We thank the Reviewer for this helpful observation. In response, we have included the ROC curves corresponding to the main predictive tasks in the Appendix, in order to visually complement the AUC values reported in the Results section.
To ensure consistency and clarity, we have also added a direct reference to these figures within the manuscript text, allowing the reader to assess the discriminative performance of the models across different thresholds.
- Q: Line 316, from where do they conclude that? Did they compare recall, precision, and F1 scores?
We thank the reviewer for this pertinent question. In our analysis, we did not use recall, precision, or F1 scores as primary evaluation metrics. Instead, we deliberately focused on the Matthews Correlation Coefficient (MCC) and Area Under the ROC Curve (AUC-ROC), which are more appropriate in the context of imbalanced clinical datasets. MCC, in particular, provides a balanced measure that accounts for all elements of the confusion matrix, even when class distributions are skewed. This choice ensures a more reliable assessment of model performance across different outcome prevalences.
To avoid any misinterpretation, we have now revised line 316 in the manuscript to more accurately reflect the metrics used and the conclusions drawn.
- Q: What is their discussion/conclusion about the SHAP figure?
A: We appreciate the Reviewer’s observation regarding the SHAP figure. The inclusion of SHAP (SHapley Additive exPlanations) was intended to enhance model transparency and provide insights into the relative contribution of each variable to the prediction of POPF risk.
To clarify this point, we have revised the Discussion to provide a more in-depth interpretation of the SHAP findings. The analysis revealed that drain fluid amylase (particularly from the left drain), serum lipase, and procalcitonin were consistently the most influential predictors of model output. Their contribution varied depending on the clinical endpoint, with DFA predominating in the prediction of POPF absence, while PCT gained relevance in the prediction of clinically relevant fistulas.
These findings not only align with current evidence regarding the pathophysiology and biomarkers of POPF, but also reinforce the clinical plausibility and reliability of the model’s behavior. We believe this strengthens the translational impact of our study and supports the use of interpretable machine learning approaches in surgical decision-making.
- Q: Their bias limitations are clear in line 337. However, they did not apply cross-validation. Bias selection of the data fold may have very high accuracy, like model AUC here, almost 1.
We thank the reviewer for raising this important point. In our study, we used a single hold-out validation split (80% training, 20% test) in order to ensure consistency across tasks and to facilitate a controlled comparison of different machine learning algorithms. This design was specifically chosen to support our primary objective: the exploration and comparison of model behavior across clinical prediction tasks, with a particular focus on explainability through XAI techniques, rather than the optimization of generalizable predictive models.
While we acknowledge that this approach does not capture variance introduced by different data partitions, we addressed this limitation by applying a bootstrap resampling procedure (1,000 iterations) on the test set. This allowed us to estimate 95% confidence intervals for AUC and MCC, providing a robust measure of statistical precision for the reported performance metrics. This methodological clarification has been incorporated into Section 2.7 of the revised manuscript.
Conclusion:
- There is no conclusion about XAI.
We appreciate the Reviewer’s observation regarding the SHAP figure. The inclusion of SHAP (SHapley Additive exPlanations) was intended to enhance model transparency and provide insights into the relative contribution of each variable to the prediction of POPF risk.
To clarify this point, we have revised the Discussion to provide a more in-depth interpretation of the SHAP findings. The analysis revealed that drain fluid amylase (particularly from the left drain), serum lipase, and procalcitonin were consistently the most influential predictors of model output. Their contribution varied depending on the clinical endpoint, with DFA predominating in the prediction of POPF absence, while PCT gained relevance in the prediction of clinically relevant fistulas.
These findings not only align with current evidence regarding the pathophysiology and biomarkers of POPF, but also reinforce the clinical plausibility and reliability of the model’s behavior. We believe this strengthens the translational impact of our study and supports the use of interpretable machine learning approaches in surgical decision-making.
To reflect this, we have added a dedicated sentence in the Conclusion to highlight the relevance of explainable AI (XAI) as a tool to improve interpretability and foster clinical integration.

Round 2
Reviewer 1 Report
Comments and Suggestions for Authors
The authors clearly addressed all the concerns—from outlining your external validation plan and hyperparameter rationale, to adding calibration/decision-curve analyses, missing-data methods, confidence intervals, feature-processing details, and a pilot protocol. With these enhancements in place, I find the manuscript ready for publication.
Reviewer 2 Report
Comments and Suggestions for Authors
The revised version submitted by the author(s) is very good and has addressed all my comments. These improvements have significantly enhanced the overall quality of the manuscript. The inclusion of ROC curve analysis and the explanation using XAI methods are appropriate and acceptable to me.